# Understanding User Practices When Drawing up Requirements—The Case of Designing Assistive Devices for Mobility

**DOI:** 10.3390/ijerph16030318

**Published:** 2019-01-24

**Authors:** Fred C. Holtkamp, Eveline J.M. Wouters, Maarten J. Verkerk

**Affiliations:** 1School for Allied Health Professions, Fontys University of Applied Sciences, P.O. Box 347, 5600 AH Eindhoven, The Netherlands; E.J.M.Wouters@uvt.nl; 2Department of Tranzo, School of Social and Behavioral Sciences, Tilburg University, P.O. Box 90153, 5000 LE Tilburg, The Netherlands; 3Department of Arts and Social Sciences, Maastricht University, P.O. Box 616, 6200 MD Maastricht, The Netherlands; maarten.verkerk@home.nl

**Keywords:** user practices, satisfaction, therapeutic footwear, rehabilitation, ankle foot orthoses, daily life of persons, Triple I model, design

## Abstract

Patients with orthopedic problems often use assistive devices, e.g., ankle-foot orthoses and therapeutic footwear, to support their mobility. However, many users are not satisfied with their devices or do not use them at all, resulting in a decrease of quality of life. It has been shown that a main cause for dissatisfaction and non-use lies in the process of drawing up requirements. It appears that orthopedic engineers have too little insight in the different areas of life of patients leading to deficient design requirements. In this article a general approach—the so-called Triple I model—is presented to understand the different areas of life of patients. This model offers, in line with and directed by the intention of the International Classification of Functioning, Disability and Health (ICF) model three perspectives or ‘pairs of lenses’ to analyze these areas of life: the identity, the interests of key stakeholders or social actors, and the underlying societal ideals. The Triple I model is elaborated for assistive devices and offers an associated methodology to orthopedic engineers to systematically map the different areas of life of patients, to understand the requirements for every area, and to explore the conditions. In case of assistive devices five different areas of life have to be investigated: daily living at home, work, transport, social and spiritual activities, sport and leisure.

## 1. Introduction

The idea of ‘quality of life’ in health care can be traced back to the definition of health by the World Health Organization (WHO) in 1948 [1]. According to WHO, health was defined in 1948 as “a state of complete physical, mental and social well-being and not merely the absence of disease or other physical defects”.

With ‘Quality of Life’ the standard of health, comfort, and happiness experienced by an individual or group is defined; in other words, it is about “the things that are needed for a good life”. Quality of Life models distinguish between utilities, achievements, and subjective evaluations and reactions [2].

The introduction of quality of life alongside health, has to be regarded as the first development to consider health from a more integral and more human perspective. This approach was hardly discussed in the world of medicine before the sixties. Only a single publication was devoted to it [3]. From the nineties on, it became a really important theme, with accompanying measuring instruments and discussions about generic and general quality of life [4].

Assistive devices, such as ankle-foot orthoses and therapeutic footwear, are prescribed and designed to improve mobility and to support physical activity, thus attributing in a positive way to quality of life of their users. For instance, Santos suggests that foot orthoses are effective in the management of patients in early stages of rheumatoid arthritis [5]. Patients may expect to see an improvement in their quality of life in three months with a further improvement in six months. This positive effect is clinically significant, provided that patients wear their orthoses for at least six months [5].

Many users of assistive devices, however, are not satisfied with their device or do not use it at all [6,7,8,9]. For example, they complain about effectiveness, comfort, dimensions, weight, and safety. Two main causes of dissatisfaction and non-use were identified [6,7,8,9]. Firstly, video- and audio recordings of intake conversations showed that orthopedic engineers when drawing up requirements, do not analyze systematically the different areas of life of their patients. Secondly, the different professionals in the orthopedic chain like doctors, orthopedic engineers, and manufacturing technicians, focus too much on their own profession, instead of jointly developing a good solution for the patient [9]. It has to be noted that the name of the various professions involved in orthopedic services in healthcare is not universal. In this article, we use the term ‘orthopedic engineer’ for professionals at bachelor and higher levels. Equivalent terms are ‘certified orthotist’, ‘orthopedic shoe technological engineer’ (orthopedic shoes), or ‘pedorthist’ (manufacturer of orthopedic shoe wear).

Many complaints of patients can be understood from the present orthopedic practice in which the daily life of the users is not systematically explored [9]. For example, patients complain that the assistive device does not support them adequately because specific movements are impossible or because it cannot be used in specific conditions, such as outdoor conditions with temperature or humidity changes. Therefore, in this paper we focus on the first cause of dissatisfaction: missing information when drawing up requirements.

In recent decades, general models such as the International Classification of Functioning Disability and Health (ICF) have been developed, to map health and environmental conditions of individuals [10]. The ICF is a framework for describing, classifying and organizing information on functioning and disability. ICF recognizes the role of environmental factors in the onset of a disease or disability, as well as the relevance of associated health conditions and their effects [10].

The ICF classification thus leads to awareness of the importance in functioning and participation, thus doing justice to the broad concept of health and quality of life. How to cover important aspects of functioning and participation in practice, is still a challenge.

For the field of orthopedic engineering, in which devices are designed to enhance functioning and participation related to mobility, the ICF cannot, as such, offer direct and practical alternatives towards possible solutions respecting the conditions in which an assistive device for mobility enhancement is used.

More specific models in the field of assistive devices are the Matching Person and Technology (MPT) model [11] and the Human Activity Assistive Technology (HAAT) [11]. The MPT model emerged from a grounded theory research and was first presented by Scherer & McKee in 1989 [11]. The MPT Model focuses on three primary areas that differentiate technology users and non-users: (a) personal and psychosocial characteristics, needs and preferences; (b) environmental factors; and (c) functions and features of the technology [12]. The HAAT model relates the personal characteristics, the objectives of the user and the assistive device that has to overcome obstacles in a certain context. The merit of the MPT and the HAAT models is that they have identified general key parameters for designing assistive devices.

However, for daily practice, a specific manner to map specific activities and environments of patients in a complete manner, is not available. The need for such an napproach a tool is apparent from earlier research that shows that orthopedic engineers do not systematically explore the different environments in which patients live and act [9]. Consequently, the design specification covers only a part of the daily life of the user with the result that only a part of the needs of the user is met, that conflicting requirements are often not made explicit, and that expectations are not sufficiently addressed [9].

The aim of this paper is to present an approach an approach for orthopedic engineers to map and to understand the needs and wishes of patients in different areas of his or her life. This development requires a broader approach that relates theory with practice, healthcare with social structures, and activities of professionals with daily life of patients. In our opinion, the so-called ‘practice approach’ fulfils this requirement [13]. Especially, we want to focus on one specific approach in this field, the so-called Triple I model, that has been developed in close cooperation with professionals in different disciplines, e.g., engineering, healthcare, and sustainability [14,15]. In this paper, we use this approach that, inspired by the ICF, can help to draw up the requirements for assistive devices.

## 2. Understanding Practices

How to understand the world of orthopedic engineers who draw up the requirements? How to understand the needs and wishes of patients? In our opinion, practice approaches are very suitable to explore the world of the orthopedic engineers and the different areas of life of patients. The main reason is that these approaches focus on the activities that are carried out in a certain context. In concrete, they concentrate on the activities done by orthopedic engineers during the intake and on the activities that patients with mobility problems would like to do in different areas of their life.

The sociologist Nicolini is also in favor of practice approaches [13]. He emphasizes that they have really something new to offer for work and organization studies [13] (pp. 1–8). One of the reasons for their usability is that they force us to rethink the role of all involved stakeholders (professionals, patients, users). Another reason is that they invite us not only to describe what people do but also to understand their activities from the perspective of meaning-making. For this reason, Nicolini believes in the transformative power of practice approaches. In his view, they are able to cause important and lasting changes in a certain field.

In this article, we would like to introduce one specific practice approach, the so-called Triple I model [11,12]. There are three reasons to present this approach. Firstly, Triple I has been developed in close cooperation with healthcare professionals and engineers. For that reason, it can be expected that it also fits to the world of the orthopedic engineer. Secondly, it unravels the complexity of practices by offering three different perspectives to understand the nature of and dynamics within a practice. Finally, it has a firm foundation in sociological, organizational and philosophical practice theories. Amongst others, it is based on the ideas of MacIntyre, Jochemsen, Glas, Hoogland and others [16,17,18,19,20].

The Triple I approach [15] offers three perspectives to understand a practice. The first ‘I’ refers to the identity and intrinsic values of a practice, the second ‘I’ to the interest of stakeholders that have a stake in the practice, and the third ‘I’ to the ideals and basic beliefs that underlie the practice, see Figure 1.

The Triple I approach is suitable to understand two types of practices. Firstly, it can be used to understand the nature of the professional practice of the orthopedic engineers who draw up requirements. This perspective is required to fathom the activities of orthopedic engineers and to comprehend why they behave as they behave. Secondly, it can be used to understand the nature of the different areas of life in which a patient uses an assistive device. This perspective is required to fathom the meaning of different activities for patients and to recognize why they act as they act in different areas of life. In this article we use the word ‘user practices’ to indicate these areas of life. We would like to emphasize that in both types of practices the behavior of individuals is understood against the background of an ongoing practice and the specific context of that practice. That means, individual orthopedic engineers and patients are not free actors but are members of social systems that stimulate or hinder certain types of behavior [13].

The Triple I model can be used in two different ways. First, it can be used in a descriptive manner to analyze an existing practice. Used as such, the model offers three perspectives or ‘three pairs of lenses’ to analyze the status quo. Second, it can be used in a prescriptive way to guide the development of an existing practice, thus, offering three perspectives to define the future status. We would like to emphasize that the three ‘I’s‘ are not isolated elements but form an integral part of a specific practice.

In this paper we will first describe the Triple I model for professional practices, focusing upon the orthopedic engineer’s practice. Second, we will use the Triple I model to understand the life of users that unfolds in different practices.

## 3. Triple I Model: Professional Practices

The Triple I model is very suitable to understand the professional practice of an orthopedic engineer, see Figure 1.

The first ‘I’ refers to the identity and intrinsic values of the practice of an orthopedic engineer. The identity of this practice can be described by ‘caring for patients with mobility problems’. In this practice technology plays a key role. After all, orthopedic engineers draft a specification for a technological device. However, that does not alter that the nature of this practice is moral: all activities in this practice have to be focused on the well-being of the patient. The intrinsic values of this practice are empathy, respect, care and trust. The relationship between identity and intrinsic values is a reciprocal one: values both concretize and constitute the identity.

The second ‘I’ refers to the interests of the stakeholders of a practice. Orthopedic engineers have their own specific network of stakeholders. For example, orthopedic doctors, insurance companies, suppliers of materials, manufacturers of assistive devices, associations of orthopedic engineers, their professional association, and so on. Every stakeholder has its own justified interests. For example, the justified interest of insurance companies is that healthcare insurance reimbursement is spent effectively. The justified interests of manufacturers of assistive devices are that the orthopedic engineers draft a good specification. And the justified interests of the association of orthopedic engineers is that their members treat patients according to the standards of the discipline.

The third ‘I’ refers to the influence of social views, cultural developments, and the spirit of the times. These phenomena are summarized with the words ‘ideals and basic beliefs’. For example, in the last decades ideals like contributing to patient autonomy and quality of life are becoming more and more important. These ideals co-shape the treatment of patients, the way of working of orthopedic engineers, and the protocols and guidelines of the practice. The third ‘I’ has a time component: it is about the ideals and basic beliefs that are embedded in the professional practice (past), that influence the professional practice (present), and that will change the professional practice (future).

First and for all, the main responsibility of orthopedic engineers is to care for their patients. In the perspective of MacIntyre [16], professionals should not just give care, but they have to excel in their profession. It goes without saying that good care in orthopedics—or: excellence in designing and manufacturing assistive devices—requires that patients are satisfied with the functioning of their device in every area of life. That implies that orthopedic engineers have to understand and analyze all the user practices of their patients.

However, the Triple I model also shows that the professional practice of orthopedic engineers is also a ‘battle scene’ of ‘actors’. This analysis helps us to understand the tensions and problems in the professional practice of the orthopedic engineer. Firstly, it invites engineers to have dialogues with stakeholders and to understand their justified interests. The orthopedic engineers have to balance the interests of the stakeholders in view of the identity and intrinsic values of the practice. Negatively formulated, orthopedic engineers are not allowed to sacrifice the intrinsic values of their practice to give in to the (unjustified) demands of stakeholders. It should be noted that some stakeholders, e.g., insurance companies, assistive device suppliers, are very demanding and easily use their power, for example, to realize price reductions. Secondly, the Triple I model makes the different interests of the various actors explicit. Ideally, this gives a clue to start a dialogue about these interests. In some cases, professionals have to convince insurance companies that from a care perspective cheap is expensive. In other cases, insurance companies have to convince professionals that a cheaper alternative also will do.

Additionally, it invites engineers to understand the ideals and basic beliefs of the present culture. These ideals and basic beliefs often challenge their professional beliefs either for good or bad. For example, ideals like ‘autonomy’ and ‘quality of life’ invite orthopedic engineers to understand the patient’s view on quality of life and to make decisions in dialogue with the patient.

In conclusion, the three I’s are ‘lenses’ to analyze and to interpret professional practices. These different lenses are not independent from each other but influence each other mutually.

## 4. Triple I Model: User Practices—The Case of Peter

Apart from understanding the professional practice of the orthopedic engineer, the Triple I model can also be used to understand the world of the patient, see Figure 1. Whereas an orthopaedic engineer works in one professional practice, the patient’s life consists of several different practices, which we call ‘user practices’. Each practice has its own identity and intrinsic values, interests, and ideals and basic beliefs, comparable to professional practices, but with a different meaning.

In the previous section we have argued that it is a primary responsibility for orthopedic engineers to understand the different areas of life of their patients. To show the strength of the Triple I model and to illustrate the practicability of this approach, we present the case study of Peter. In the next section we show that this case is useful to develop a general methodology.

Peter is 58 years of age, married and without children. Peter and his wife have a dog. They live in a small apartment without an elevator in the city center at the second floor. He works as a maintenance technician in a large factory, specialized in the manufacturing of microchip-controlled machinery. He commutes each working day between his residence and his employment by bus. In his spare time, he goes out fishing, preferably at sea, and one evening per week he is practicing yoga. Peter has had a back operation due to a hernia. During this operation one of the nerves that controls the foot was damaged. Since that time, Peter suffers from a drop foot. That means, he has an impaired ability to raise his toes and raise the foot from the ankle. As a result, all walking movements are more difficult. Especially, when the surface is uneven and in case of complex movements. Also, the risk of falling is increased. To overcome these limitations, he needs an assistive device, i.e., an ankle foot orthosis. The short description of the life of Peter shows that five areas of life are important and involve mobility: daily life at home, work, transport, social activities and spirituality, and sports and leisure. In all these areas of life the assistive device has to support the mobility of Peter. It has to be noted that these categories are the most important ones for assistive devices [9], see Figure 2. Successively, we will analyze these five user practices.

(1) Daily life at home. The identity of this practice is social, it is about living together with his wife. The most important intrinsic values are being together, supporting each other and enjoying life. The most important stakeholder or social actor in this user practice is the wife of Peter. Her main interest is that Peter feels happy in his house, can move freely in and around the house and is able to cook independently, because her working hours are later than Peter’s. However, it is difficult to maneuver in their small kitchen. In addition, the bathroom is also very small. Another important aspect is that Peter always has to use the stairs to enter or to leave his home because the building has no elevator. This analysis shows that the assistive device has to support Peter in moving in small rooms and in using the stairs. Peter has been raised up in a traditional family. A large part of the household is done by Peter’s wife. However, under influence of the ideals and basic beliefs of the present time they believe that Peter has to function autonomously and independently as much as possible. That means, the assistive device has to support Peter in performing all activities in the household. Including activities like cleaning that require movements like stooping and kneeling down.

(2) Work. The identity of this practice is technical and economic. Important intrinsic values of this practice are the quality of the maintenance activities and the safety of the equipment. The most important stakeholders or actors in this practice are his boss and his colleagues. The main interest of his boss is that Peter performs his activities in an efficient manner and that his drop foot does not increase his absenteeism. The main interest of his colleagues is that Peter is flexible about his work in case of absence of one or more of his colleagues. That means, the assistive device has to support Peter in all movements that are needed for maintenance, e.g., sitting on his knees, crawling under machines, standing on stairs, and moving and turning in small spaces. The maintenance practice functions in the context of the whole company. One specific ideal and basic belief come to the fore. The company policy is ‘safety first’. That means, Peter’s assistive device needs to fit into safety shoes. Finally, the working conditions are excellent. Peter works in rooms that are temperature and humidity controlled.

(3) Transport. The identity of this practice is movement between places. For example, home-work traffic, visiting family and friends, shopping in the city center. Its most important values are convenience and safety. The most important means of transportation for Peter are bus, train and bicycle. The interest of the owner of the bus and train company: Peter has to be able to get in and out easily and quickly. The interest of fellow passengers is that they can travel with as little as possible barriers and hindrance. Peter uses the bus for travelling to work. In the bus (which is crowded at rush hours) he has to be able to take small steps and also to walk sideways. He also uses his bike for transport in the city center. In biking he has to push the pedals (circular motion) and to move his leg sideways when he is stationary. For the assistive device this means that sufficient support needs to be delivered when biking and therefore the device needs to be constructed with a certain stiffness. Important ideals and basic beliefs in the field of transport are about sustainability. However, this will not influence Peter’s life strongly, nor the design of his device.

(4) Social activities and spirituality. The identity of the Yoga practice is mainly spiritual: it is about being meaningful for other people and searching for inner peace. For Peter the intrinsic values of yoga are a perfect combination of finding peace and tranquility. In addition, Yoga implies doing exercises that help to stay in a fit shape and gain more self-consciousness. It is a fact that some of the exercises cannot be performed due to the rigidity of the orthosis Peter is wearing. The main interest of the Yoga teacher is that Peter can do his exercises without hindering his fellow participants.

(5) Sport and leisure. The identity of the practice of fishing with friends at sea is a typical social activity. Main values are friendship, being together and enjoying life. Peter wants to be a very close friend. The most important stakeholders are his friends. They want to support Peter with carrying the materials and casting the rods. However, Peter feels himself a bit ashamed that he needs an assistive device. He wants to do all activities himself as much as possible. He wants to participate in fishing as a ‘normal’ person. The assistive device should especially help him to make small steps in the boat and to adjust his balance in case of heavy waves. The assistive device also has to stand the conditions of sea fishing, e.g., salt, moisture and cold. Finally, the most important ideals and basic beliefs in fishing have to do with sustainability. However, that will have no influence on the specification of the assistive device.

Table 1 gives a summary of all details of the life of Peter that are relevant for prescribing an assistive device. This information provides the orthopedic engineer with the relevant know how to discuss the best possible solution with the user, to make together a decision, and to draw up detailed specifications. For example, Peter needs for his work to sit on his knees, crawl under machines and move and turn in small spaces. On top of that he also needs to wear safety shoes at work. In the case of Peter an assistive device in the form of an ankle foot orthosis seems suitable to support the ankle and foot. For the design it should be considered that a certain rigidity from the support is desired but that the ankle foot orthosis is also flexible and small enough to fit into his safety shoe without pinching the toes. In relation to the specification this means that the orthopedic engineer needs to think over how these demands are to be realized, for example using a stiff material in a construction that allows a certain flexibility or a flexible material with some construction elements that stiffens the ankle foot orthosis in certain places. An alternative solution can be a custom-made orthopedic safety shoe to use during his work, in combination with an ankle foot orthosis and confection shoes to use in his home and leisure time.

In conclusion, the case of Peter clearly shows the fruitfulness of the Triple I model in analyzing the life of Peter. It offers a practical approach to consider the different areas of life systematically. It gives the opportunity to discuss the variety of possible solutions and to choose the preferred solution together with the user in a process of shared decision making.

We would like to highlight two elements of this analysis. Namely, that the nature or the character of the various user practices are fundamentally different. This also implies that the idea of meaning-making is different in these practices. 

The meaning of daily life is related to the social aspect (living together) and moral aspect (caring for each other). The meaning of work is often related to the economic aspect (economic contribution to society, earning money). The meaning of yoga is related to the spiritual aspect (finding peace). The idea that different practices have different meanings is based on the theory of individuality structures as developed by the philosopher Dooyeweerd [14,20]. Table 2 gives an overview of examples of different natures of user practices. The second element we would like to highlight is that the case of Peter clearly shows that the Triple I model addresses the personal and environmental factors of the ICF model in-depth. The personal factors especially come to the fore in the first I: the intrinsic values from the perspective of the patient and the meaning-making aspect and the environmental factors in the second I: the stakeholders.

## 5. Developing A Methodology to Draw up Specifications

The case of Peter shows that patients use their device in different areas of life, and that every area has its own requirements, needs and conditions. The methodology behind this case is summarized in Figure 3.

The first step is to identify the different practices in which a patient will use the assistive device.

The second step is to execute a Triple I analysis for every user practice. It should be noted that such an analysis might be time consuming. However, this investment is very important to understand the needs and wishes of the patient. Furthermore, during the patient examination time is available to explore these practices.

The third step is to perform an activities-movements analysis for every practice. That means: for every practice the activities of the patient have to be mapped and the associated movements have to be made explicit.

The fourth step is to specify the conditions for every practice. The most important parameters are the condition of the surface, and physical conditions like temperature and humidity.

The last step is to draw up the specification. Beforehand, however, it cannot be guaranteed that all needs and wishes can be realized in one device. For example, in certain cases descending the stairs asks for a relatively flexible way of moving the foot. For walking, a more immobilized foot to generate sufficient support is needed. In this case, these two conflicting demands, flexibility and rigidity, can be realized in the device by a partly ridged and flexible construction around the foot.

Additionally, it is possible that it is difficult to realize some needs or wishes from out of a cost-effective or a constructional point of view. For example, a particular person would like to ride the bike but has insufficient control of the foot. In this case a standard assistive device, for example an ankle foot orthosis, cannot accomplish all wishes and demands. In this example the variety of demands needs to be discussed with the user to decide for the most optimal solution or in case there is no optimal solution, the expectations of the patient have to be managed.

## 6. Discussion

The Triple I model offers an approach to map the different contexts in which the user acts and the assistive device will be used, the specific movements that will be made within each practice, and the particular conditions of a practice that have to be considered. The Triple I model and the associated methodology can be interpreted as a realization of The International Classification of Functioning, Disability and Health (ICF), a broadly accepted and widely used system to improve the quality of life of patients and citizens [21,22]. This classification focuses on personal and environmental factors, activities and participation. The personal factors include gender, age, coping styles, social background, education, profession, past and current experience, and behavior. The environmental factors include social attitudes, architectural characteristics, and legal and social structures, as well as climate, terrain, and so forth. We would like to emphasize that the ICF was a basic concept in the application of the Triple I model for the concept of user practices.

The Triple I model and the associated methodology offer a concrete method to analyze the personal and environmental factors as well as the performed activities and participation for every area of life in which the assistive device is used.

The MPT model [11] focuses on personal and psychosocial characteristics, needs and preferences, environmental factors, and functions and features of the technology. The HAAT model [12] evaluates the personal characteristics, the objectives of the user and the assistive device to overcome obstacles in a certain context. Both models identify some key parameters for designing assistive devices and are as such, comparable to the Triple I model. However, the Triple I model distinguishes itself from these models in four ways. First, it recognizes that patients live in various practices and that each practice has its own needs and conditions. Second, it emphasizes that every practice has its own nature and character: different types of meaning making. In the words of Nicolini [13], practices are not only about facts but also about ‘meaning-forming’ (p. 7). This goes beyond the objective descriptive nature of ICF, MPT and HAAT. Third, it offers a methodology to explore systematically and thoroughly all relevant user practices of an individual patient. Finally, it is sensitive for changing societal opinions about health, disabilities and diseases, as it includes the perspective of ‘Ideals and basic beliefs’.

The Triple I approach will induce a change in the professional practice. Presently, orthopedic engineers do not investigate the areas of life of patients in-depth [9]. In our view, the main reason is that the ICF model does not present a detailed methodology to do so. The Triple I model and its methodology, however, urges the orthopedic engineer to analyze all areas of life in which the patient has to use the assistive device and has to understand the meaning-making of all these areas from the personal characteristics of the patient. We believe that this change will lead to a transformation in the way the professional is acting: only in this way the patient with his or her preferences, motivations, needs and wishes, will become the focus of what is called ‘patient centered’ design and care.

To find its way in research and clinical practice, the Triple I model and its methodology have to be translated in, e.g., a flowchart model and an (automated) structured questionnaire (internet application). In this way the user concerned can provide information on their practices in a systematic way to the orthopedic engineer. This could be already performed at home before the consultation. During the intake or patient examination process these answers are to be discussed. On first sight, the process of asking and answering questions might seem time consuming. However, the advantage is that much more information becomes available in a systematic way to be used in the design process, leading to better products, better alignment to expectations and shared decision making, less adjustments afterwards, and less second opinions. And that, on its turn, saves a lot of time.

Remarkably, the Triple I model does not only fit the present views on health, it also reflects the, currently often cited, ideas of health of Hippocrates. In antiquity, health was considered to be a state of balance [22]. Hippocrates stated that it is ‘more important to know what person has a disease than to know what sort of disease a person has’ [23]. From this perspective, the Triple I model can be understood as an approach to explore ‘what person has a disease’ and how this person can ‘adapt and self-manage’ with respect to orthopedic challenges.

This article has some limitations. In application of the Triple I model we only focused on assistive devices in the orthopedic domain. However, it is believed that this approach will be useful in other mobility related design processes as well, both technological and service designs. In addition, it is believed that this approach can be applied to the design of all medical devices that patients use in different areas of life [15,17,19]. Finally, although thoroughly theory based, the model needs to be empirically tested in practice. For further research, we therefore recommend to develop a practical tool as suggested for orthopedic engineers and provide training, and evaluate, amongst others, the satisfaction of patients after implementation of the tool.

## 7. Conclusions

The Triple I model examines the user practices of patients. This model supports orthopedic engineers in understanding the life of their patients. It raises awareness that the assistive device will be used in different life areas and that each area poses its demands. We have presented a methodology for every life area or user practice in a systematic way in order to make an inventory of the expectations, needs and demands with respect to assistive devices. The application of the Triple I model and its associated methodology will increase the quality of the specification, which is a precondition to improve the quality of orthopedic care and to increase the quality of life of patients with mobility problems. It has to be noted that extensive mapping of the different life areas may reveal that it is not possible to meet all the requirements within one device. At the same time, the model helps to clarify such impossibilities, and helps to discuss and address expectations with the user. Further research needs to be undertaken in to examine the consequences of mapping the user practices for the design requirement specification set up.

## Figures and Tables

**Figure 1 ijerph-16-00318-f001:**
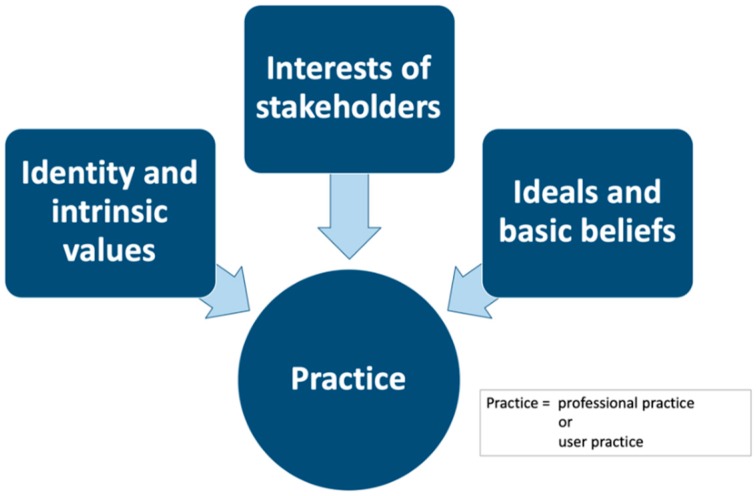
Graphical representation of the Triple-I model for practices. (A practice can be professional practices or user practices).

**Figure 2 ijerph-16-00318-f002:**
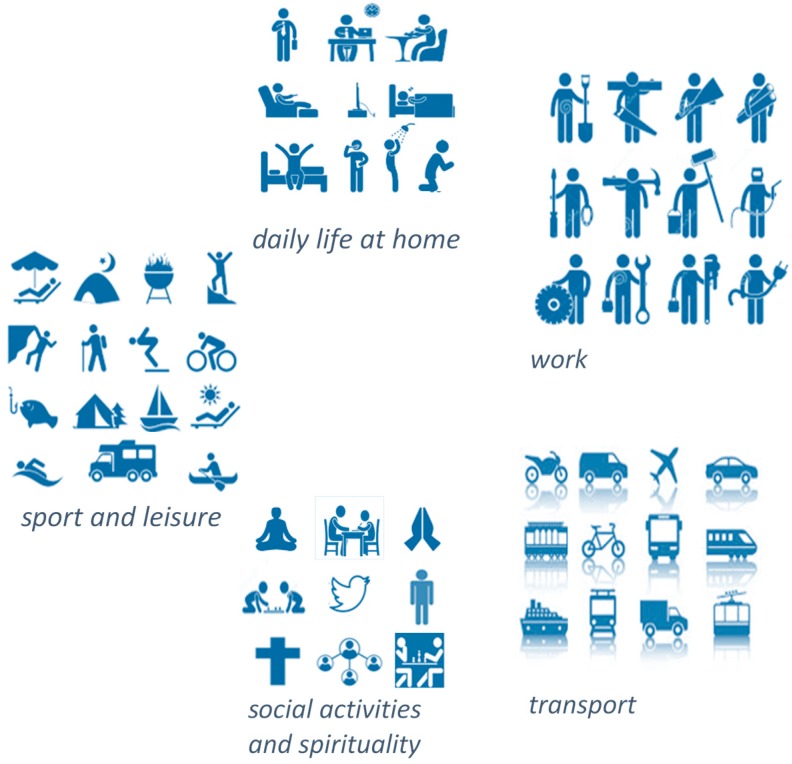
The five user practices; (1) daily life at home, (2) work, (3) transport, (4) social activities and spirituality, and (5) sport and leisure.

**Figure 3 ijerph-16-00318-f003:**
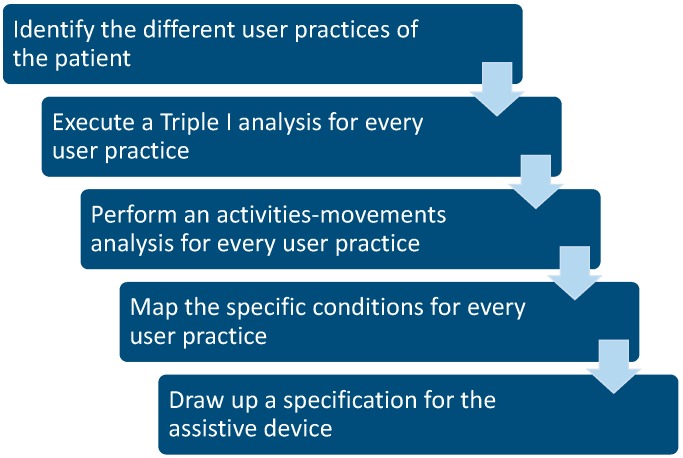
Graphical representation of the Triple-I methodology for user practices to set up the specifications for the assistive device.

**Table 1 ijerph-16-00318-t001:** A summary of all details of the life of Peter that are relevant for prescribing an ankle foot orthosis.

	User Practice or Areas of Life	I1	I2	I3	Specific Movements	Conditions
		Identity and Intrinsic Values	Interests of Stakeholders	Ideals and Basic Beliefs		
1	Daily life at home	Social context: living—being together, supporting each other, enjoying life	Spouse is main stakeholder: her main interest is that her husband is happy and can move freely in and around the house	Freedom, autonomy, independency	Maneuvering in small rooms (kitchen), walking stairs, walking in and outside the house	Standard
2	Work	Technical and economic context: quality and safety	Employer is main stakeholder: his interest is that Peter works in an efficient way	Self-development	Sitting on his knees, crawling under machines, standing on stairs	Wearing safety shoes
3	Transport	Transport context: convenience and safety	Bus and train company are main stakeholders: biking in public space, all users of public space are stakeholder: get on and off easily, no hindrance for fellow passengers and public space users.	Social and economic participation, independency	All movements inherent to buses, trains and bicycles (short step, sideway steps, circular motion etc.)	Standard, moisture (rain)
4	Social activities and spirituality	Social and spiritual—being together, being meaningful, finding inner peace	Teacher and fellow participants are main stakeholders. Their main interest is not hindering fellows and independent practicing the exercises.	Independency, self-consciousness, independency	Yoga exercises	Wearing no shoes—barefoot
5	Sport and leisure	Social context: friendship, being a very close friends	Friends are most important stakeholders: they want to get along easily with all activities	Social participation	Small side steps in the boat, a good balance in case of large waves	Salt, moisture, coldness

**Table 2 ijerph-16-00318-t002:** Overview of different natures of user practices from the perspective of meaning-making.

Category	User Practice	Nature
Daily life at home	Daily life at home	Social (living together)
		Moral (caring for each other)
Work	Work in industry	Economic (contribution to society, livelihood)
	Work in shops	Economic (contribution to society, livelihood)
	Work in health care	Moral (caring for patients), economic (livelihood)
	Work as a musician or artist	Esthetic (beauty), economic (livelihood)
Transport	Transport for work	Economic (part of work)
	Transport for social activities	Social (to meet other people)
Social activities and spirituality	Political activities	Juridical (making good laws)
	Religious activities/Yoga	Spiritual (inner peace)
	Visiting museums	Esthetic (enjoying beauty)
	Visiting a concert	Esthetic (enjoying beauty)
	Member board of foundation	Depends on the type of the foundation
Sport and leisure	Soccer, handball, athletics and so on	Physical (exercise), social (doing together)
	Chess, checkers and so on	Intellectual (thinking), social (doing together)
	Cooking	Physical (activity), social (enjoying family and friends), esthetic (enjoying taste)
	Cycling, walking and so on	Physical (activity), social (enjoying accompany)

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
