# Peer review of "Understanding User Practices When Drawing up Requirements—The Case of Designing Assistive Devices for Mobility"

_ijerph, 2019, doi:10.3390/ijerph16030318_

Round 1

Reviewer 1 Report

When drawing up requirements it is indeed very important to unsterstand both professional and user practices since all these factors (may) influence benefit of assistive devices. Therefore studying these factors is an important issue. The authors point towards and have used an interesting model that reflects several of these factors. Both research and clinical practice may benefit from making use of the Triple-I model. Table 1 for example is very useful and shows the prospect of the study. However, the present article has some important flaws that would need editing:

The introduction briefly starts with concepts as positive health and quality of life, but the clear link with the rest of the manuscript is not very strong. What is the influence of that 'there are no specific tools to investigate...' (line 77) on both concepts? Also, the introduction needs major restructuring, given that several sentences or parts of paragraphs are repeated. For example lines 108 and further resemble line 123. The introduction doesn't really help the reader towards the main problem in practice. 

Only briefly in the discussion the ICF is mentioned, although concepts as activity and participation seems to be highly linked to the issue of drawing up requirements. These need to be imbedded both in the introduction as well as in the discussion of the Triple I model and case description.

Are Figure 2 and 3 really needed? They closely resemble figure 1 and seem to be redundant. Although 2 and 3 look at different perspectives in the practice circle, I would suggest adding these to figure 1. Also, it seems to me that there are close links and bidirectional relations between 'identity and intrinsic values', 'interests of stakeholders', and 'ideals and basic beliefs'. How do these influence each other?

It is not clear from the manuscript how precisely the factors influence the professional practice. I think the manuscript and the prepared case study would benefit from describing how the professional practice is influenced in the case of Peter as well. How does the professional practice informs clinical decision making next to the user practice? There are several perspectives that should lead to the drawing up of requirements and weighing these in the light of professional practice, Insurance etc. Especially because authors state in line 294 'difficult to realise' and 'expectations of the patient have to be managed' (line 297). Where does this fit in the model for clinical decision making?

Peter suffers from a drop foot. To be able to really show the benefit of the model, please think about describing how the drop foot currently influences Peter on activity and participation level. 

In line 270 authors state that this analysis might be time consuming. There is no reflection on this in the discussion. If it is time consuming, then what is the added value for clinical practice? What should you do to decrease the time? Perhaps drawing up some sort of decision making tool? How would the model find its way to research and clinical practice? This should be reflected upon in the discussion, in my opinion. The discussion should better reflect possibilities and possible problems of integrating Triple I in both research and clinical practice. 

Author Response

First of all, we would like to thank the reviewers for their time and effort put in the review of the article.  We are pleased with the fact that all reviewers emphasized the value of the model.Your suggestions and comments have provided us with valuable insights and have led to a tightening of the model and improvement of the article.

Reviewers' comments:

When drawing up requirements it is indeed very important to understand both professional and user practices since all these factors (may) influence benefit of assistive devices. Therefore, studying these factors is an important issue. The authors point towards and have used an interesting model that reflects several of these factors. Both research and clinical practice may benefit from making use of the Triple-I model. Table 1 for example is very useful and shows the prospect of the study. However, the present article has some important flaws that would need editing:

Authors: Thank you very much for these friendly words and compliments and your suggestions for improvement.

The introduction briefly starts with concepts as positive health and quality of life, but the clear link with the rest of the manuscript is not very strong. What is the influence of that 'there are no specific tools to investigate...' (line 77) on both concepts? The link between Qol and the triple I in service of the setup of the design requirements is explained in line 80.:  as a consequence…..”

Authors: Thank you very much for this remark. We agree with your recommendation, therefore, the text in the introduction has thoroughly rewritten.

From the original file (lines 35 – 43) are deleted.

Authors: we also have deleted lines 59 and 60. This has been done in order to focus more on the essence of this article. 

In lines 62 to 68, we now emphasised the relationship between quality of life and the objective of this article.:  

“Assistive devices such as AFO’s …. Attributing in a positive way to quality of life of their users… " 

To make the link clearer we have deleted the lines (line 79-101) 

Currently, there are no specific tools that take their starting point in the daily life of users and that explore the different areas of life of orthopedic patients systematically in order to draw up the specifications for an assistive device.  

We also added text in 

line 68 “orthopedic services in healthcare”

line 76-77 “missing information when drawing up requirements.”

Also, the introduction needs major restructuring, given that several sentences or parts of paragraphs are repeated. For example, lines 108 and further resemble line 123. The introduction doesn't really help the reader towards the main problem in practice. 

Authors: Thank you very much for this recommendation. We agree with you and as already mentioned we have restructured the introduction. We now start with explaining what the definition of Quality of life is. Then we elaborate on this In relation to assistive devices in the spectrum of mobility, focus on what is achieved and what is missed in our opinion. We then continue with the explanation of the MPT and HAAT models also suggested by the reviewer.

Authors: We added the lines 84-92 

“More specific models in the field of assistive devices are the Matching Person and Technology (MPT) model [11] and the Human Activity Assistive Technology (HAAT) [11].  The MPT model emerged from a grounded theory research and was first presented by Scherer & McKee in 1989 [11]. The MPT Model focuses on three primary areas that differentiate technology users and non-users: (a) personal and psychosocial characteristics, needs and preferences; (b) environmental factors; and (c) functions and features of the technology [12]. The HAAT model relates the personal characteristics, the objectives of the user and the assistive device that has to overcome obstacles in a certain context. The merit of the MPT and the HAAT models is that they have identified general key parameters for designing assistive devices.” 

Authors: In our opinion the original lines 108 were explaining and elaborating the triple I model. Lines 123 and further are explaining how the triple I model can be used.   Therefore, in our opinion, it is not a repetition but a further explanation of triple I use.However, your comments forced the authors to rethink and rewrite this section.

Authors: the introduction section with the added text below (lines 101-108)

"The aim of this paper is to present a tool for orthopedic engineers to map and to understand the needs and wishes of patients in different areas of life. The development of such a tool requires a broader approach that relates theory with practice, healthcare with social structures, and activities of professionals with daily life of patients. In our opinion, the so-called ‘practice approach’ fulfils this requirement [13]. Especially, we want to focus on one specific approach in this field, the so-called Triple I model, that has been developed in close cooperation with professionals in different disciplines, e.g., engineering, health care, and sustainability [14-15]. In this paper, we use this approach to develop a tool for drawing up the requirements for assistive devices."

As reviewer 2 suggested; We changed the word “spectacles” in pair of lenses according your recommendation, in line 21, and line 215-216 

Only briefly in the discussion the ICF is mentioned, although concepts as activity and participation seems to be highly linked to the issue of drawing up requirements. These need to be imbedded both in the introduction as well as in the discussion of the Triple I model and case description.

Authors: Thank you very much for this note and the recommendation for embedding this. We agree with your recommendation and changed the text accordingly.

This has been added in the introduction in lines 77-83 the case description and discussion (377-380).

“In recent decades, general models such as the International Classification of Functioning Disability and Health (ICF) have been developed, to map health and environmental conditions of individuals [11]. The ICF is a framework for describing, classifying and organizing information on functioning and disability. Although the ICF recognizes the role of environmental factors in the onset of a disease or disability, as well as the relevance of associated health conditions and their effects [11], it is designed to classify these circumstances. It is not offering alternatives towards possible solutions respecting the conditions in which an AD for mobility enhancement is used.” 

As well as in the discussion: lines 377-380.

“The Triple I model and the associated methodology can be interpreted as a realization of The International Classification of Functioning, Disability and Health (ICF), a broadly accepted and widely used system to improve the quality of life of patients and citizens [21-22].”

Are Figure 2 and 3 really needed? They closely resemble figure 1 and seem to be redundant. Although 2 and 3 look at different perspectives in the practice circle, I would suggest adding these to figure 1. 

Authors: Thank you very much for this comment. We agree with this recommendation. The figures 2 and 3 are removed accordingly. The original figures together with figure 1 are now combined to one (1) figure, figure 1. Figure 1 has given an extra text frame explaining how practice can be interpreted.

Authors: Respectively, the original figures 4 and 5 are now Figure 2 and 3, However the sequence of these figures have been changed due to changes in the text

Also, it seems to me that there are close links and bidirectional relations between 'identity and intrinsic values', 'interests of stakeholders', and 'ideals and basic beliefs'. How do these influence each other?

Authors: Thank you very much for this comment. To explain more about the relation between Identity, Interests and Stakeholders we have changed the original sentence (lines 168-170) into: 

“The intrinsic values of this practice are empathy, respect, care and trust. The relationship between identity and intrinsic values is a reciprocal one. On the one hand, values concretize the identity of the practice, and on the other hand, values constitute the identity”. 

Authors: As mentioned in the manuscript, “the relationship between identity and intrinsic values is a reciprocal one. On the one hand, values concretise the identity of the practice, and on the other hand, values constitute the identity".  Identity and intrinsic values, interests an ideal are really about different axes: intrinsic values are at a different level (for many there is a reason to work in health care) while the interests (of insurers, for example), during daily work is a relic that can be at odds with that, but ultimately does not influence that identity (but can cause tension). Idem about ideals.And of course, there is a close link / relationship between the values and their influences.  As stated already in the manuscript: “We would like to emphasise that the three ‘I’s‘ are not isolated elements but form an integral part of a specific practice”.  

It is not clear from the manuscript how precisely the factors influence the professional practice. I think the manuscript and the prepared case study would benefit from describing how the professional practice is influenced in the case of Peter as well. 

Authors: Thank you very much for your suggestion for improvement. 

In answer to the question, "how precisely the factors influence the professional practice” it is very hard to reply with a general answer. We stated in an earlier published article (Holtkamp F.C. Verkerk M.J. Van Hoof J, Wouters E.J.M. Mapping user activities and user environments during the client intake and examination phase: An exploratory study from the perspective of ankle foot orthosis users. Technol Disabil. 2016, 28, 145-157, DOI: 10.3233/TAD-160452), The normative practice approach appears to be very fruitful as it unravels the complexity of practices, identifies the normative aspect of these practices, and highlights the influence of basic beliefs, world views, and religions.” We are now in the stage of knowing that all these factors exist and have an influence. 

In the discussion (lines 402-405) we added: “The Triple I approach will induce a change in the professional practice as shown by the case of Peter and the followed methodology. This change could lead to a transformation in the way the professional is acting: only in this way the patient with his or her preferences, motivations, needs and wishes, will become the focus of what is often called ‘patient centered’ design and care. “

Authors: At this moment it is too early to determine precisely what the influence is. Further research is needed.

How does the professional practice inform clinical decision making next to the user practice? 

Authors:  An explorative study in health care. (Phil Ref 2017, 82920 167-171.  DOI: 10.1163/23528230-08202001) shows that the performance of health care will strongly improve when care is organized around the patient instead of around the professional. That health care performance will further improve when dialogues are organized to handle the layered structures of identity and intrinsic values, and ideals and basic beliefs. Thirdly, the idea of user practices offers a new means to understand the daily life of patients and to make an inventory of their needs and wishes.

In the text (lines 140-146) the text is changed in: 

“The Triple I approach is suitable to understand two types of practices. Firstly, it can be used to understand the nature of the professional practice of the orthopedic engineers who draw up requirements. This perspective is required to fathom the activities of orthopedic engineers and to comprehend why they behave as they behave. Secondly, it can be used to understand the nature of the different areas of life in which a patient uses an assistive device. This perspective is required to fathom the meaning of different activities for patients and to recognize why they act as they act in different areas of life. In this article we use the word ‘user practices’ to indicate these areas of life.”

            Authors:  Line (365) is changed: 

“Additionally, it is possible that it is difficult to realize some needs or wishes from out of a cost-effective or a constructional point of view.”

Authors: Not further elaborated upon in this article, but one can imagine that during the process of investigating the needs and wishes of a client by examining the identity and intrinsic values, ideals and basic beliefs etc. one or more specific wishes and the resulting requirements are contradictory to each other. As a result, one can conclude that such a solution that will overcome certain limitations is not possible to manufacture.  That outcome must be discussed with the client. Firstly, to inform the client about the (im)possibilities and secondly, to manage the expectations. 

Authors: in our conclusion (lines 437-439) we therefore stated:  

“The Triple I model starts with the life of the use and how he lives his life by examining the user practice. This model may support orthopedic engineers in understanding the life of their patients. It raises awareness that the assistive device will be used in different life areas and that each area makes its own demands. We have presented a methodology for every life area or user practice in a systematic way in order to make an inventory of the expectations, needs and demands with respect to the assistive devices. The application of the Triple I model and its associated methodology will increase the quality of the specification, what is a precondition to improve the quality of orthopedic care and to increase the quality of life of patients with mobility problems.”

Peter suffers from a drop foot. To be able to really show the benefit of the model, please think about describing how the drop foot currently influences Peter on activity and participation level. 

Authors, thank you very much for this recommendation and we agree with you. The text in the manuscript (lines 230-234) has been adapted accordingly: 

"Peter suffers from a drop foot. That means, he has an impaired ability to raise his toes and raise the foot from the ankle. As a result, all walking movements are more difficult. Especially, when the surface is uneven and in case of complex movements. Also, the risk of falling is increased. To overcome these limitations, he needs an assistive device, i.e., an ankle foot orthosis.”

In line 270 authors state that this analysis might be time consuming. There is no reflection on this in the discussion. If it is time consuming, then what is the added value for clinical practice? What should you do to decrease the time? Perhaps drawing up some sort of decision making tool? How would the model find its way to research and clinical practice? This should be reflected upon in the discussion, in my opinion. The discussion should better reflect possibilities and possible problems of integrating Triple I in both research and clinical practice. 

Authors, thank you very much for your comment. To reflect on this statement, we have added in the discussion the text in lines 460-463:  

“During the intake or patient examination process these answers are to be discussed.  On first sight, the process of asking and answering questions might seem time consuming. However, the advantage is that much more information becomes available in a systematic way to be used in the design process, leading to better products, less adjustments afterwards, and less second opinions. And that, on its turn, saves a lot of time.”

Your recommendation, or question to draw up some sort of decision-making tool, is of course an ambitious objective that we are would like to develop in the near future. But to do so, we first need to have insight in the information gathering and requirement drawing process. Therefore, we need insights obtained using the triple I model. The decision-making tool is a next step.

Authors: in order to address and explain this, we also have added the following text in the discussion (lines 407-411):

“To find its way in research and clinical practice, the Triple I model and its methodology has to be translated in e.g., a flowchart model and an (automated) structured questionnaire (internet application). In this way the user concerned can provide information on their practices in a systematic way to the orthopedic engineer. This could be already performed at home before the consultation. During the intake or patient examination process these answers are to be discussed.”

Reviewer 2 Report

This is an interesting commentary on an understudied area of rehab. I agree with the premise and the need to improve this AT delivery process. But I am not sure this is a research paper (they state “This research has some limitations”), rather it is an opinion/commentary (via fictional example) about how a specific model can be applied to AT prescription. No research was performed, no methods/protocol, etc. I suggest this be rewritten to emphasize this distinction.

No evidence of this conclusion is provided “The Triple I model supports orthopaedic engineers in understanding the life of their patients.” It *may* support this, but no research has been done to show this.

The authors have presented a very detailed example of what things one might consider when specifying a device. But why not end this with an example of the actual specification, instead of just all the issues that still remain, e.g. 280 “it cannot be guaranteed that all needs and wishes can be realised in one device”? It is a bit troubling that even in this hand-picked fictional example the 3-I method does not produce a definitive spec? And there is no mention of how one might deal with conflicting needs/requirements. Should the model not help guide the practitioner to a solution, and one that most practitioners would independently come up with? More on the Requirements (that is given such prominence in the title).

161  “First and for all, this analysis shows that the main responsibility of orthopaedic engineers is to care for their patients.” I am not sure how your analysis showed this (but I do agree). Moreover, the 3-I model in the Figs implies that all 3 I’s are equally impactful. Why not have a hierarchical representation if the 1st I is indeed the main one.

Much space is devoted to QoL and other high level ideas about society and health care. None is devoted to other models that might better (or not) fit AT delivery. What about AT models like HAAT, MPT, etc. A complete treatment of this topic should discuss alternatives, pros/cons, etc.

197 description of the life of Peter shows that four [*five*] areas of life are important

Table 1 is the center piece of this paper, the most interesting info. But should another column be added about the requirement/spec (e.g. stiffness) that arises from each domain?

In the example I would like to see more emphasis on the tech requirements (e.g. 235 “device needs to be constructed with a certain stiffness”).

138 “In this practice technology plays a key role” – you do somewhat reconcile the tech issues/stakeholders, for instance, about funding constraints. But is still sounds like professional practice “ideals” assumes any and all tech is available and funded (but just maybe funders will grind you on price). But in practice they are more highly constrained – some products might not be available or suitable (or more likely simply not acknowledged to the client since the salesperson/dealer may not like a certain manufacturer or may not like the margin they receive), there might not be enough funding, there may be pressure to consider certain products due to higher margins. Maybe more can be said about this ‘battle scene’ of ‘actors’ and how it impacts good outcomes.

I suggest instead of “pairs of glasses” you say “lenses” or “filters”, which is often used like this in English.

I think Figs 1-3 can be just 1 figure. Not much new info is being added with the other 2 figs.

Author Response

Response

Title: Understanding User Practices, When Drawing up Requirements—The Case of Designing Assistive Devices for Mobility

First of all, we would like to thank the reviewers for their time and effort put in the review of the article.  We are pleased with the fact that all reviewers emphasized the value of the model.Your suggestions and comments have provided us with valuable insights and have led to a tightening of the model and improvement of the article.

Reviewers' comments:

This is an interesting commentary on an understudied area of rehab. I agree with the premise and the need to improve this AT delivery process. 

Authors: Thank you very much for these friendly words and compliments.  

But I am not sure this is a research paper (they state “This research has some limitations”), rather it is an opinion/commentary (via fictional example) about how a specific model can be applied to AT prescription. No research was performed, no methods/protocol, etc. I suggest this be rewritten to emphasize this distinction.

Authors: Thank you very much for this comment and we fully agree with you. Indeed, this article is a clinical note - viewpoint article. However, to avoid any confusion we have change this in the text by using only the word ‘article” we removed the adjective.

No evidence of this conclusion is provided “The Triple I model supports orthopaedic engineers in understanding the life of their patients.” It *may* support this, but no research has been done to show this.

Authors: Thank you very much for this comment. According you comments we changed the text and the first sentence (line 437) is rewritten into: 

“The Triple I model may support orthopedic engineers in understanding the life of their patients.” 

At the end of the conclusion the following text is added (lines 447-448): “Further research needs to be undertaken in to examine the understanding of the user practices in the design requirement specification set up.”

The authors have presented a very detailed example of what things one might consider when specifying a device. But why not end this with an example of the actual specification, instead of just all the issues that still remain, e.g. 280 “it cannot be guaranteed that all needs and wishes can be realised in one device”? It is a bit troubling that even in this hand-picked fictional example the 3-I method does not produce a definitive spec? And there is no mention of how one might deal with conflicting needs/requirements. Should the model not help guide the practitioner to a solution, and one that most practitioners would independently come up with? More on the Requirements (that is given such prominence in the title).

161  “First and for all, this analysis shows that the main responsibility of orthopaedic engineers is to care for their patients.” I am not sure how your analysis showed this (but I do agree). 

Authors: we removed the part “this analysis shows that”..  The sentence  (line 187) now is rewritten into: 

“First and for all, the main responsibility of orthopedic engineers is to care for their patients.”

Authors: we have chosen not to end with an example of a possible actual specification because in our opinion part of the specification should be more than just for instance the material specifications. Also, constructive specifications should be part of this. Although our article is about Understanding User Practiceswhendrawing up specification, the focus is on the user practices as a a starting point to formulate demands and wishes. To end with a full specification, materials and constructive, is a next step.

Moreover, the 3-I model in the Figs implies that all 3 I’s are equally impactful. Why not have a hierarchical representation if the 1st I is indeed the main one.

Authors: regarding the figures and suggestion that these have a hierarchical representation we would like to clarify that the triple I model has no hierarchical meaning for the three I’s’. All I’s’ are equally important. Therefore, we have not altered this. 

instead of just all the issues that still remain, e.g. 280 “it cannot be guaranteed that all needs and wishes can be realized in one device”? It is a bit troubling that even in this hand-picked fictional example the 3-I method does not produce a definitive spec?

Authors: in reply on your comment regarding line 280 in the original text – now line 360 “it cannot be guaranteed that all needs and wishes……..confection shoes to use in his home and leisure time.”

Authors; Therefore, even the development of this triple I model does not guarantee that all wishes and needs can be granted. But the limitations will be clear and must be clarified for patients. 

Authors: In the discussion is referred to (lines 402-405)

The Triple I approach will induce a change in the professional practice as shown by the case of Peter and the followed methodology. This change could lead to a transformation in the way the professional is acting: only in this way the patient with his or her preferences, motivations, needs and wishes, will become the focus of what is called ‘patient centered’ design and care. 

The triple I model therefore helps by making a thorough and complete overview of the practices of the user and the meaning of these practices for the user.  The complete overview of all answer and thus information results in a well-informed orthopaedic engineer who can conduct in a good conversation with the patient, submit choices and make the right decisions.

Much space is devoted to QoL and other high level ideas about society and health care. None is devoted to other models that might better (or not) fit AT delivery. What about AT models like HAAT, MPT, etc. A complete treatment of this topic should discuss alternatives, pros/cons, etc.

The description of the life of Peter shows that four [*five*] areas of life are important

Authors: Thank you very much for this remark. Indeed five areas of life are important. (the word ‘four” was a mistake in the original text) We have change this accordingly, (line 234)

Authors: The MPT and HAAT model are now part of this article in the introduction and discussion. 

“The Human Activity Assistive Technology model (HAAT) model as well as the Matching Persons and Technology (MPT) model are added in the text (Lines 84-92) .More specific models in the field of assistive devices are the Matching Person and Technology (MPT) model [12] and the Human Activity Assistive Technology (HAAT) [13].  The MPT model emerged from a grounded theory research and was first presented by Scherer & McKee in 1989 [12]. The MPT Model focuses on three primary areas known to most differentiate technology users and non-users: (a) personal and psychosocial characteristics, needs and preferences; (b) milieu/ environmental factors; and (c) functions and features of the technology being evaluated [13]. The HAAT model relates the personal characteristics, the objectives of the user and the assistive device that has to overcome obstacles in a certain context. The merit of the MPT and the HAAT model is that they have identified some key parameters for designing assistive devices.  

   (lines 428-437) Both the Matching Person and Technology model, that focuses on personal and psychosocial characteristics, needs and preferences, environmental factors, and functions and features of the technology being evaluated and the Human Activity Assistive Technology model that the personal characteristics, the objectives of the user and the assistive device to overcome obstacles in a certain context relates, have identified some key parameters for designing assistive devices and these do have comparable elements with the Triple I model. However, the tipple I model starts with the user practices and is in an explicit way sensitive for personal preferences and motivation which are directly related to a practice. Nicolini emphasizes that it is not only about facts but also meaning making. In other words: it is about what the patient thinks is important. This is something that as well as  ICF, MPT and HATE not offer.”

Table 1 is the center piece of this paper, the most interesting info. But should another column be added about the requirement/spec (e.g. stiffness) that arises from each domain? In the example I would like to see more emphasis on the tech requirements (e.g. 235 “device needs to be constructed with a certain stiffness”).

Authors: thank you very much for this comment and question. Another column can be added, such as material specifications, e.g., stiffness.  However, from the scope of this article we would like to focus on the practices. A column with all technical specification is the ultimate objective, and will be the next step of our research. We will consider, apart from, e.g., stiffness, also other material and then also construction specifications of interest with respect to the design. To do so, we believe that first a full understanding of the user practice is of importance (the scope of this article).  The derivate of this is the consequence for the requirements in detail. 

Authors: For the sake of clarity, we have included a second table, which is now Table 2, that combines the user practices with the nature of a practice. This from the viewpoint of meaning-making of the user, to answer the question, what is of importance for the user?

138 “In this practice technology plays a key role” – you do somewhat reconcile the tech issues/stakeholders, for instance, about funding constraints. But is still sounds like professional practice “ideals” assumes any and all tech is available and funded (but just maybe funders will grind you on price). But in practice they are more highly constrained – some products might not be available or suitable (or more likely simply not acknowledged to the client since the salesperson/dealer may not like a certain manufacturer or may not like the margin they receive), there might not be enough funding, there may be pressure to consider certain products due to higher margins. Maybe more can be said about this ‘battle scene’ of ‘actors’ and how it impacts good outcomes.

Authors: We thank you very much for this comment and we fully agree on the fact that there is much more to elaborate on this issue. However, within the scope of this article “understanding the user practices.” in our opinion it is too much to start and elaborate this “battle scene” discussion, next to the establishment of the awareness of the practice theory, the technical specifications etc. Perhaps we may conclude in a very precocious way, that introducing the practice theory (from out of the professional practice theory) an awareness is created for all these issues and practices.

I suggest instead of “pairs of glasses” you say “lenses” or “filters”, which is often used like this in English.

Authors:  Thank you for this suggestion. In line (152-153) the text is changed accordingly.

“Used as such, the model offers three perspectives or ‘three pairs of lenses’ to analyze the status quo.”

I think Figs 1-3 can be just 1 figure. Not much new info is being added with the other 2 figs.

Authors: Thank you very much for your suggestion.

We removed the figures in 2 and 3. These figures are combined with figure 1 to one (1) figure, (figure 1). Figure 1 has been given an extra text frame explaining how practices can be interpreted.

As a consequence, the original figures 4 and 5 are now Figure 2 and 3.

Reviewer 3 Report

This is an important paper as user centered design is critical to successful use of AT

The model is very useful

pg 1 3

line 35-37 cna you clarify what you mean here 96-98 You mention Nicolinis practice approaches can you describe this a bit more 

Author Response

Rebuttal

Title: Understanding User Practices, When Drawing up Requirements—The Case of Designing Assistive Devices for Mobility

First of all, we would like to thank the reviewers for their time and effort put in the review of the article.  We are pleased with the fact that all reviewers emphasized the value of the model.Your suggestions and comments have provided us with valuable insights and have led to a tightening of the model and improvement of the article.

Reviewers' comments:

This is an important paper as user centered design is critical to successful use of AT

The model is very useful

Authors: Thank you very much for these friendly words and compliments.  

Line 35-37 can you clarify what you mean here 96-98 You mention Nicolinis’ practice approaches can you describe this a bit more 

To explain what Nicolini meant with his practice approach we are referring to his book “Practice Theory, Work, & Organization, an introduction”, would like to explains that a practice-bases view of social and human phenomena is distinctive in that it:

·Emphasizes that behind all apparently durable features of our world,.. , there is some type of productive and reproductive work.

·Forces us to rethink the role of agents and individuals (manager, the managed, etc.)

·Foregrounds the importance of the body and objects in social affairs.

·Sheds new lights on the nature of knowledge and discourse

·Reaffirms the centrality of interest and power in everything we do.

Authors: We adapted the text to make clearer that because of adopting the practice approaches the orthopaedic engineer also needs to think over wat is important for the user, what is the meaning making of the different user practices. And therefore, what is most important.

Authors: we changed the text into:

 “Second, it emphasizes that every practice has its own nature and character: different types of meaning making. In the words of Nicolini [13], practices are not only about facts but also about 'meaning-forming' (p. 7). This goes beyond the objective descriptive nature of ICF, MPT and HAAT.”

Round 2

Reviewer 1 Report

After revision the manuscript has been improved on major parts. Especially the introduction benefits from adding ICF, MPT and HAAT, making a stronger case for this study and working with the triple I model.

Still, there are some (minor) comments or questions to make, in my opinion:

Although described in the introduction and discussion, ICF could play a more distinctive role in this study, especially in the description of the case. Why not use ICF for qualifying the severity of impact on activity and participation level? It would benefit the manuscript if authors could show how several models and framework altogether are used and be integrated in decision making. Please also look at personal and environmental facilitators. These would make a difference in decision making, although two patients may experience somewhat the same limitations in body functions or structure.

It remains unclear how professional practices are influenced. The answer from the author feels very unsatisfying, in only that "we are now knowing factors exist and have influence", "will enduce change..." and "transformation in the way professional is acting". This is very vaguely described, although this should and would be one of the major values of the triple I model: hoe to influence professional practice. If we still are not sure enough about how professional practice can or will be influenced by using this way of drawing up requirements, then is there really an added value to the model? Authors need to make a stronger case.

Author Response

Dear Reviewer,

Thank you very much for your second review and compliments. We are happy that you consider our revision as an improvement of the manuscript. Also, we appreciate your in-depth question about the use of the ICF. 

In your review to wrote: “Although described in the introduction and discussion, ICF could play a more distinctive role in this study, especially in the description of the case. Why not use ICF for qualifying the severity of impact on activity and participation level? It would benefit the manuscript if authors could show how several models and framework altogether are used and be integrated in decision making. Please also look at personal and environmental facilitators. These would make a difference in decision making, although two patients may experience somewhat the same limitations in body functions or structure.”

Authors: When we introduced the triple I model, it has never been, or will be our intention to neglect the ICF. Triple I should and would be used to classify or qualify the severity of impact on activity and participation level, which is addressed in the ICF model. It is our intention to introduce the triple I model as a tool to operationalize ICF in practice for persons who have an assistive device prescribed. 

Authors: We changed the text in line 334 ‘two elements’. 

Authors: We added the following text in lines (336-340)

The second element we would like to highlight is that the case of Peter clearly shows that the Triple I model addresses the personal and environmental factors of the ICF model in-depth. The personal factors especially come to the fore in the first I: the intrinsic values from the perspective of the patient and the meaning-making aspect. The environmental factors in the second I: the stakeholders.

Authors: In the discussion we have added the text below (lines 398) to make the relation between ICF and Triple I clearer. 

…. and so forth. We would like to emphasize that the ICF was a basic concept in the application of the Triple I model for the concept of user practices. 

Authors: We changed the summary (from line 28 onward) into: This model offers, in line with and directed by the intention of the International Classification of Functioning, Disability and Health (ICF)….

Authors: In line 81 (In the original text) we removed “Although the.

Authors: The text is now: ICF recognizes the role of environmental factors in the onset of a disease or disability, as well as the relevance of associated health conditions and their effects [10].

and removed in line 82 also:  “it is designed to classify…’

Authors: We changed line 124 in this paper into: “in this paper, we use this approach to develop a tool, that, inspired by the ICF, can help to draw up the requirements for assistive devices. 

Reviewer: It remains unclear how professional practices are influenced. The answer from the author feels very unsatisfying, in only that "we are now knowing factors exist and have influence", "will induce change..." and "transformation in the way professional is acting". This is very vaguely described, although this should and would be one of the major values of the triple I model: hoe to influence professional practice. If we still are not sure enough about how professional practice can or will be influenced by using this way of drawing up requirements, then is there really an added value to the model? Authors need to make a stronger case.

Authors: We would like to thank the reviewer for the critical remark on professional practices and we agree that it is important to explain this more clearly in our manuscript. Therefore, we added extra text in the discussion paragraph. 

Authors: We added the lines (409-417) as stated below: 

The Triple I approach will induce a change in the professional practice. Presently, orthopedic engineers do not investigate the areas of life of patients in-depth [9]. In our view, the main reason is that the ICF model does not present a detailed methodology to do so. The Triple I model and its methodology, however, urges the orthopedic engineer to analyze all areas of life in which the patient has to use the assistive device and has to understand the meaning-making of all these areas from the personal characteristics of the patient. We believe that this change will lead to a transformation in the way the professional is acting: only in this way the patient with his or her preferences, motivations, needs and wishes, will become the focus of what is called ‘patient centered’ design and care.

Reviewer 2 Report

They have addressed most of the concerns I raised. A few minor english/typos remain. 

Author Response

Dear reviewer,

Authors: Thank you very much for your second review and compliments. We are happy that you consider our revision as an improvement of the manuscript.

They have addressed most of the concerns I raised. A few minor english/typos remain.

Authors: ‘We have asked a colleague with high English proficiency to proof read the entire manuscript in a ‘fresh’ manner and were able to change some typo’s. We would like to thank the reviewer to make us aware of these typo’s.
